# The Reliability and Accuracy of a Fall Risk Assessment Procedure Using Mobile Smartphone Sensors Compared with a Physiological Profile Assessment

**DOI:** 10.3390/s23146567

**Published:** 2023-07-20

**Authors:** José-Francisco Pedrero-Sánchez, Helios De-Rosario-Martínez, Enrique Medina-Ripoll, David Garrido-Jaén, Pilar Serra-Añó, Sara Mollà-Casanova, Juan López-Pascual

**Affiliations:** 1Instituto de Biomecánica (IBV), Universitat Politècnica de València, Edificio 9C, Camino de Vera S/N, 46022 Valencia, Spain; jose.pedrero@ibv.org (J.-F.P.-S.); helios.derosario@ibv.org (H.D.-R.-M.); enrique.medina@ibv.org (E.M.-R.); david.garrido@ibv.org (D.G.-J.); juan.lopez@ibv.org (J.L.-P.); 2Unidad de Biomecánica Clínica (UBIC), Department of Physiotherapy, Faculty of Physiotherapy, Universitat de València, Carrer Gascó Oliag 5, 46010 Valencia, Spain; pilar.serra@uv.es

**Keywords:** fall risk, smartphone, inertial sensors, Physiological Profile Assessment, smartphone, inertial sensors, Timed Up and Go

## Abstract

Falls in older people are a major health concern as the leading cause of disability and the second most common cause of accidental death. We developed a rapid fall risk assessment based on a combination of physical performance measurements made with an inertial sensor embedded in a smartphone. This study aimed to evaluate and validate the reliability and accuracy of an easy-to-use smartphone fall risk assessment by comparing it with the Physiological Profile Assessment (PPA) results. Sixty-five participants older than 55 performed a variation of the Timed Up and Go test using smartphone sensors. Balance and gait parameters were calculated, and their reliability was assessed by the (ICC) and compared with the PPAs. Since the PPA allows classification into six levels of fall risk, the data obtained from the smartphone assessment were categorised into six equivalent levels using different parametric and nonparametric classifier models with neural networks. The F1 score and geometric mean of each model were also calculated. All selected parameters showed ICCs around 0.9. The best classifier, in terms of accuracy, was the nonparametric mixed input data model with a 100% success rate in the classification category. In conclusion, fall risk can be reliably assessed using a simple, fast smartphone protocol that allows accurate fall risk classification among older people and can be a useful screening tool in clinical settings.

## 1. Introduction

Falls in older people are a significant health concern since their incidence is higher than 30% in people older than 65 and twice as high in people over the age of 80 [1]. In fact, falls are the leading cause of disability in older people and the second most common cause of accidental death [2]. The early detection of fall risk is crucial for prevention; thus, many fall risk assessment methods and tools have been developed in the last few decades. Most are based on single or combined observations derived from questionnaires and functional performance measurements.

Questionnaires are used to evaluate risk factors such as previous falls, a fear of falling, physical or cognitive issues, and comorbidities [3] and usually result in a classification often reduced to binary discrimination between “fallers” and “nonfallers”. However, fall risk may be more accurately modelled as a continuum with fuzzy boundaries between multiple risk categories [4].

Performance measurements objectively quantify the capacity to respond to physical and cognitive demands, whose decline is directly related to the risk of falling during daily activities. These associations have been demonstrated in gait disturbances, body balance, lower limb strength, and reaction time [5,6]. Tests such as the Timed Up and Go (TUG) [7], sit-to-stand (STS) [8], the Tinneti balance scale [9], the Berg balance scale [10], the Short Physical Performance Battery (SPPB) [11], and the Physiological Profile Assessment (PPA) [12] have shown an excellent ability to predict falls and physical function in older people [13]. 

Simple tests are more appropriate for clinical settings because of the practitioner’s workload. However, they do not always provide a general overview of the clinical problems, and those using combined measurements can be time consuming [14]. This could explain why regular screenings are performed by only one-quarter of general practitioners [15] and could motivate further research, leading to fall risk assessment techniques that overcome implementation barriers, such as duration and complexity.

Sensor technology may reduce the burden of measuring, and over the last few years, many instrumented versions of performance assessments have been developed and tested, especially on wearable devices [16]. The main advantage of instrumented protocols is that the results are less dependent on the interpretation of the evaluators, as already shown by Weiss and colleagues, who concluded that reliability increased from 63% to 87% [17] when the TUG was assessed in an instrumented procedure.

Previous studies have investigated the effectiveness of different sophisticated systems designed to assess fall risk in older adults. These systems (i.e., pulse-Doppler radar, Kinect, ultrasound, time-of-flight sensors, and web cameras [18]), individually or in combination, can even be installed in the user’s house, providing unobtrusive, continuous quantitative activity and gait assessment. The data collected using this technology can provide a reliable balance and functional assessment that the clinician or researcher can use to calculate a patient’s fall risk. Thus, this technology offers a great advantage for older adults’ fall detection, e.g., with regard to fall risk assessment and its real-time changes [19,20].

However, although noteworthy results have been obtained using this type of technology, transferring these procedures to clinical practice for general population assessments has been difficult because of the challenge of continuously monitoring users [18], which requires intensive work in the clinical context, where human resources are usually scarce. Therefore, accurate, easy-to-use, and affordable fall risk prediction approaches usable in the clinical setting are needed. For this purpose, other types of sensors have also been investigated, such as force, pressure, and inertial sensors, for use on an ad hoc basis in the clinical setting. An advantage of the latter is that it can be used in instrumented functional assessment tests, such as the TUG, without requiring large facilities or complex calibrations.

This study presents a new inertial sensor-based fall risk assessment protocol in which a combination of physical performance measurements is continually recorded with an inertial sensor embedded in a smartphone. The aim of this protocol was to evaluate the reliability of this new fall risk assessment method and its validity by comparing its results to the results obtained by Physiological Profile Assessment (PPA). This protocol gathers information equivalent to various functional tests with minimum complexity and time consumption. It includes measurements based on key factors related to fall risk, such as balance, reaction time, gait, and lower limb strength. 

Furthermore, we aimed to determine the most accurate classification model for predicting different fall risk levels by comparing the accuracy of parametric and nonparametric models using neural networks and raw signal data [21]. These models extract features automatically and detect the relevant characteristics in the classification process. The results were compared with the classification scoring obtained by the PPA and a battery of physical and sensory tests based on vision, reaction time, leg strength, proprioception, and balance. These have been proven valid and reliable tools that can predict the risk of falls and are widely used in research and clinical practice [12].

## 2. Materials and Methods

### 2.1. Participants

The assessments were conducted at the Health Centre of Alcudia (Valencia, Spain). All participants provided written informed consent, and the procedures were performed in accordance with the principles of the World Medical Association’s Declaration of Helsinki. The Experimental Research Ethics Committee of the Universitat Politècnica de València approved the protocol (P4211016). 

The inclusion criteria were 55 years and older and the ability to walk at least 10 metres without any walking aid. The exclusion criteria were the presence of motor alterations after neurological disorders that interfered with mobility, severe uncorrected visual or auditory disorders preventing the tests from being conducted, and inability to understand instructions (Mini-Mental State Examination < 23 points).

Sixty-five participants were recruited for the study, and they were rated as “very low fall risk”, “low fall risk”, “mild fall risk”, “moderate fall risk”, “high fall risk”, and “very high fall risk” according to their PPA score. They were also rated as “low” (values below 1 or “low fall risk”) or “high” (above 1 or “high fall risk”) according to their PPA score [22] for a higher sample size in each group. A priori sample size calculation was performed using a type I error of 0.05 and a power of 80%, taking into account “medial – lateral displacement” outcome results from previous studies and including two groups (i.e., fallers and nonfallers) [23].

### 2.2. Experimental Procedure

The mobility assessment was performed using the FallSkip^®^ application (Biomechanical Institute of Valencia, Valencia, Spain), a Java App running on a smartphone (Xiaomi Redmi 4 x Model MAG138). The app processes the data recorded at 100 Hz by the smartphone-embedded 3D inertial sensor (High-Performance 6-Axis MEMS Motion TrackingTM) composed of a 3-axis gyroscope, 3-axis accelerometer, and Digital Motion ProcessorTM (TDK-ICM-20689; TDK, Tokyo, Japan). The smartphone was attached with a hook-and-loop fastener on a strap around the waist of the participants, just below the iliac crest point near the centre of mass (Figure 1). Furthermore, previous studies have determined that the kinematics of the lower lumbar region provide information on energy consumption during gait [24].

An application was used to capture the raw data from the mobile sensors and to guide the researcher throughout the assessment using on-screen indications and audio signals. The materials required to conduct the test (a variation of the TUG test) were a 3-metre free aisle and an armless chair and were validated in previous studies [25,26,27]. The participants were placed standing up 3 metres away from the chair. The instructions to perform the test were:t0. Stand with trunk straight and arms along the body for 30 s (t0);t1. When an audio signal is emitted by the smartphone (30 s after the beginning of the test), walk straight at a normal pace toward the chair (t2–t1);t2. Stop for 2 s in front of the chair;t3. Turn and sit down (t4–t3);t4. Wait for 2 s;t5. Start to stand up;t6. Stand up (t6–t5);t7. Walk straight at a normal pace to the starting point (t8–t7);t8. Finish the test.

Each participant repeated the test 3 times, with a resting time of 1 min between repetitions. After the three repetitions, a trained physiotherapist assessed the participants’ fall risk using the PPA.

### 2.3. Sensor Data Analysis

Data from the smartphone-embedded sensors were collected at a fixed sampling rate of 100 HZ, and the raw signals were analysed as in a previous study by our research group [22,26,27,28]. An interpolation of the recorded signal was performed to ensure that the data points were evenly distributed in time and that no information was missing between consecutive samples [28], and a fourth-order Butterworth low-pass filter with zero lag at 20 Hz was applied. Finally, the test phases described below were identified using characteristic local peaks of the accelerometer and orientation signals.

Calculating the dependent variables began by determining the sensor’s orientation from the accelerations and angular velocities using Favre’s method [29]. These orientations, expressed in Euler angles (Roll, Pitch, and Yaw), were used during the segmentation process of the test phases (balance, gait, turn to sit, sit to stand, and gait) by identifying the instants where changes of direction (turn to sit) occurred.

The sensor position was calculated analytically in the frequency domain by the double integration of the acceleration signal using the Fourier transform and its inverse, as in [30], to avoid cumulative drift caused by the integration process. 

The following balance, gait, and functionality parameters were extracted from the position signals in different phases of the test: (i) Anterior–posterior and medial–lateral displacement of the centre of masse (CoM) during the 30 s standing (**APDisp, MLDisp**), calculated as 90th percentile of the double integration of the accelerometer signal and an inverted pendulum model [31]; (ii) Vertical and medial–lateral excursion of CoM while walking (t2–t1 and t8–t7) (**VRange, MLRange**), calculated in the same way as in [32]; (iii) Average power of turning–sitting movements (t4–t3) and standing up (t6–t5) (**PTurnSit, PStand**), estimated by the trajectory of the CoM, the weight of the participant, and the time it takes for stand-to-sit and sit-to-stand [33]; (iv) Range of anterior–posterior jerk of CoM during turning–sitting movement and standing up (**APJerkSit, APJerkStand**); (v) Reaction time (**Reaction_Time**), measured as the time elapsed between the ringing of the audio signal and detecting walking motion (t1–t0); (vi) Total motion time (**Total_Time**), computed as the sum of the split of walking (t2–t1 and t8–t7) and sit (t4–t3) to stand times (t6–t5).

### 2.4. Statistical Analysis

The analysis was conducted using the R package for statistical computing in RStudio (R Foundation for Statistical Computing, version 3.5.3). Classical statistical methods were used to obtain the mean as a measure of central tendency and the standard deviation (SD) as a measure of dispersion. Before carrying out the inferential analysis, we checked for extreme values and the assumptions of normality (using the Shapiro–Wilk test) and homoscedasticity (using the Levene test). In the case of noncompliance with any of the assumptions, nonparametric tests were considered.

For the inferential analysis, the intraclass correlation coefficient for the average of random measures (ICC)(2,1) was calculated to assess the reliability of the eight test variables. The results were averaged over the different repetitions for each subject. Moreover, a between-groups Student *t*-test was performed to determine whether there were differences between the groups (fallers and nonfallers) on the variables cited above (APDisp, MLDisp, VRange, MLRange, PTurnSit, PStand, APJerkSit, APJerkStand, APJerkSit, APJerkStand, Reaction_Time, and Total_Time). The type I error was set at 5% (*p* ≤ 0.05), and the power was set at 0.8.

### 2.5. Classification Models

Different classification models were generated from parametric models that used the abovementioned variables and from nonparametric models that directly used the raw signals for the classification. Two versions were generated from all models, the first with two levels of fall risk classification (faller and nonfaller) [22] and the second that was classified into six levels of fall risk (very low (level 1), low (level 2), mild (level 3), moderate (level 4), high (level 5), and very high (level 6)), according to the risk levels defined by the PPA [12].

#### 2.5.1. Parametric Models

All parametric models we compared were available in the Sklearn library [34] for Python 3.7.x [35]. To process and prepare the input data, a pipeline was generated by (i) normalising the variables using the StandardScaler, which transforms the data to a mean equal to 0 and an SD of 1 [34], (ii) recursive feature elimination (RFE), which recursively selects the variables with the best classification results (in this step, we established a maximum value for the six parameters because we had few observations for the high number of predictors [36]), and (iii) by implementing the models of Logistic Regression, Ridge Classifier, LASSO, K Neighbours, Gaussian Naive Bayes, Linear Discriminant Analysis, Decision Tree, Perceptron, Multilayer perceptron, Stochastic Gradient Descent, Gradient Boost, XGBoost, Support Vector Machine, Random Forest, and AdaBoost.

#### 2.5.2. Nonparametric Models

The nonparametric model was a multibranch neural network (Figure 2), with the raw sensor signals in the time domain as the first input branch and the spectrogram image (frequency domain) of the signals as the second input branch.

For the first input branch (i.e., raw signals), a 64-sample moving window of the accelerometer and the gyroscope was conducted, with an input data structure of 64x6 being the three channels of each sensor (x, y, z). The second input branch (i.e., the spectrogram), a short-time Fourier transform (STFT) provided by the TensorFlow 2.9.1 framework, was performed on the windows that entered the model through the first branch. To extract the frequency information of each of the six signals individually, all the signals were combined into a single signal of 384 samples (6 signals × 64 samples). A short-time Fourier transform (STFT) was then performed on this new signal using frame length = 20 and frame step = 2 to create a signal that was as square as possible. In order to enhance the visibility of amplitudes, the logarithm of the absolute values obtained from the STFT function was taken.

As indicated above, the method involved two input branches. The first branch utilised the raw signals’ sliding windows. The features were extracted through a series of 1D convolutional and dropout layer concatenations with ReLu activation functions and 64, 128, 512, and 1024 filters, respectively. These features were then passed through two long- and short-term memory (LSTM) layers to obtain the sequential properties of the signals. Three dense layers with ReLu activation functions were concatenated with the other branch.

The second branch used the spectrogram image of the signals. Three 2D convolutional and dropout layers with a kernel size of 3 × 3 and activation function ReLu were concatenated.

The top model employed two dense layers with 128 and 64 neurons with a Relu activation function. The output layer used two or six neurons (depending on the model version), one for each class, with a softmax activation, to obtain the percentage of belonging to each class.

To compile the model, categorical cross-entropy was used as the loss measure and the Adam optimiser. The evaluation metric “accuracy” was specified, and a batch size of 32 was used for 100 training epochs to fit the model. The classification model was developed using Keras API and Tensor Flow 2.0 in Python 3.7.x. The methodology presented in [37,38,39] inspired this approach.

We employed a grid search methodology to systematically investigate various hyperparameter combinations, including learning rate, batch size, and number of epochs. The model’s performance was assessed on the training and validation sets of each experiment. By adjusting the hyperparameters based on the experiment results, we iteratively repeated the process until reaching the optimal performance.

#### 2.5.3. Validation and Comparison Models

The sample of 195 measurements (65 participants × 3 repetitions) was split into one set for training and validation and another for testing the models, using an 80/20 stratified proportion. To fit the models, a repeated stratified *k*-fold was performed with 10 splits and 10 replicates for a total of 100 cross validations.

To compare the results of different models, we calculated (i) the accuracy during training, (ii) the mean accuracy of the stratified 10-fold cross validation, (iii) the accuracy during testing, (iv) the geometric mean of the test sample, and (v) the F1-score as the harmonic mean of precision and recall.

## 3. Results

### 3.1. Participants

Sixty-five participants were recruited for the study. Their baseline characteristics are presented in Table 1, and the differences between fall risk groups were consistent with known fall risk factors (gender, age, obesity, and history of falls).

The distribution and reproducibility of the variables used in the model are summarised in Table 1. All variables had ICC values between 0.74 and 0.93. Considering that normality and homoscedasticity were assumed, *t*-tests were performed to compare the faller and nonfaller groups, and significant differences ($p{\;}_{value}<0.05$) were found between groups for all variables.

### 3.2. Comparison of Classification Models

The results of all implemented models are shown in Table 2. Most of the parametric classification models with two levels of risk had an average accuracy of around 0.65 with test data, decreasing to 0.6 when classifying the six fall risk levels. This suggests that our model shows more possibilities to classify well into two than into six levels. The proposed neural network-based classifier showed an accuracy of 0.99 and 0.98, respectively.

**Table 1 sensors-23-06567-t001:** Comparison of the variables calculated in the functional test and the reliability between the groups.

Variables	Nonfallers (*n* = 40) Mean (SD)	Fallers (*n* = 25) Mean (SD)	Total (*n* = 65) Mean (SD)	*t*-Test *p*-Value	ICC
Age	66.75 (6.87)	71.44 (6.83)	68.55 (7.18)	0.009	-
Weight (kg)	74.84 (10.73)	77.97 (15.57)	76.04 (12.78)	0.34	-
Height (m)	1.64 (0.08)	1.58 (0.08)	1.62 (0.09)	0.016	-
MLDisp (mm)	5.92 (2.34)	8.36 (4.73)	6.86 (3.63)	0.007	0.84
APDisp (mm)	16.88 (4.53)	21.12 (11.53)	18.51 (8.17)	0.041	0.74
Vrange (mm)	29.19 (4.77)	22.62 (5.31)	26.66 (5.90)	<0.001	0.76
Mlrange (mm)	54.94 (17.08)	45.17 (15.18)	51.18 (16.94)	0.022	0.93
PTurnSit (W)	133.25 (44.21)	102.56 (31.35)	121.45 (42.26)	0.004	0.91
Pstand (W)	281.58 (137.77)	220.03 (68.05)	257.91 (119.22)	0.042	0.35
APJerkSit (m/s^3^)	18.79 (5.15)	15.39 (4.52)	17.49 (5.16)	0.009	0.85
APJerkStand (m/s^3^)	22.60 (5.89)	19.12 (6.42)	21.26 (6.28)	0.029	0.82
Reaction_Time (s)	0.82 (0.40)	1.21 (0.52)	0.97 (0.49)	0.001	0.93
Total_Time (s)	12.82 (2.39)	14.77 (3.44)	13.57 (2.97)	0.009	0.86

ICC—Intraclass Correlation Index of the three repetitions of the test; MLDisp—range of the medial–lateral displacement of the centre of mass (CoM) in balance phase; APDisp—range of the anterior–posterior displacement of the CoM in balance phase; VRange—range of vertical displacement of the CoM in gait phase; MLRange—range of medial–lateral displacement of the CoM in gait phase; PTurnSit—power of turn to sit; PStand—power to sit to stand; APJerkSi—anterior–posterior jerk to sit; APJerkStand—anterior–posterior jerk to sit and to stand.

## 4. Discussion

In this study, we proposed a new method to assess the risk of falling in healthy older adults using an inertial sensor embedded in a smartphone. The method was designed taking into consideration the main requirements for use in clinical settings, the simplicity and short-term administration, feasibility for older people, and portability [12,25,26,27]. The validity and reliability have also been assessed to ensure its utility in fall risk detection. The method developed was based on measurements widely used in other fall risk assessments, such as balance, gait quality, reaction time to initiate gait, and time required to sit and get up from a chair, performed in a unique test similar to the TUG [13,22].

The test–retest reliability was fair to excellent for all variables computed from the assessment, with ICC ranging from 0.75 to 0.93. These ICC values were generally higher than the ICCs obtained from the reliability assessment of the different tests included in the PPA, whose ICC ranged between 0.50 and 0.97 [12]. When comparing our procedure with the original TUG widely used in this population, we obtained similar ICCs for the Total_Time variable and the time required to perform the complete TUG in community-dwelling adults (ICC between 0.916 and 0.960) [40].

To validate the ability of the assessment procedure to classify fall risk severity, we developed parametric and nonparametric models and assessed the accuracy of fall risk classification. As our results show, parametric models, in which computed variables were included, are appropriate for assessing fall risk in older people since they showed high accuracy in classifying fall risk levels in the sample. Nevertheless, when the results of the parametric models were compared with the nonparametric model using neural networks, the qualitative leap in accuracy was quite relevant. Parametric models obtained an average accuracy of 0.7, while the nonparametric one exceeded 0.9, regardless of being applied to the dichotomous classification (i.e., fallers and nonfallers) or polytomous classification with six fall risk categories corresponding to the PPA. In addition, it is worth noting that achieving correct outcomes with parametric models is comparatively simpler when selecting from two possibilities (50% chance for fallers and nonfallers) than from 1/6 probability of being correct (17% chance for PPA levels).

These findings are similar to those obtained in previous research published by our research group [21], which has already compared different types of classification models. This current study demonstrates that a simple instrumented procedure and a neural networks analysis of the raw data could provide objective, reliable, and valid assessment, avoiding the intrinsic problems of more complex laboratory assessments (i.e., long-lasting, expensive, and complex evaluations). It could be argued that using another technology or combining different devices for data recording (for example, having more than one inertial sensor) could increase the precision of the clinical variables measured by a smartphone. However, this may impact the usability and speed of assessments, which are essential in a clinical context.

While many previous studies concentrated on evaluating fall risk by analysing gait data rather than functional activities [41,42], recent studies have used smartphone applications to assess fall risk [43]. Nevertheless, the present study brings an innovative vision to these classifiers, as it not only puts them into two customary classifications but also divides them into six categories. This can allow clinicians to have a rapid, easy-to-use, and reliable fall risk classification and, therefore, create a prevention programme to avoid future falls.

When our accuracy results were compared with previous studies in which wearable sensors were used to classify fall risk groups, similar results were obtained [44,45,46] but using a less time-consuming protocol. A recent study has demonstrated that neural network analysis is able to classify older adults as fallers and nonfallers using sensor data based on the TUG test. Contrary to our study, their sample included different health conditions, such as cognitive and physical impairments, and was thus more heterogeneous than ours. Therefore, although the method followed in this study included safeguards to avoid overfitting the model, the size and diversity of the sample might be insufficient to generalise the results. However, it is a good starting point for future studies with a larger sample size, including fall risk classification for specific pathologies, as in previous studies [25,26].

In addition, the developed model allowed combining patient data and the sensor’s raw signal data, providing a combination of raw data in the time and frequency domains. Including an additional input branch to the model that considers the frequency aspect of the signals may lead to an increase in accuracy of up to 99% and 98% for identifying two or six fall risk levels, respectively. Future studies with larger samples may also test the inclusion of the clinical variables as a third independent branch of the model since this approach has shown very promising accuracy for classifying Parkinson’s disease stages [47].

The proposed model provides six levels of fall risk classification aligned with the PPA [12]. PPA administration takes approximately 40 min since it includes various tests involving vision, reaction time, leg strength, proprioception, and balance domains. In contrast, the protocol proposed in this study using the inertial sensor device includes elements of all these assessments embedded in a functional test (i.e., modified TUG). This could enable clinicians to perform the evaluation in 105 s on average, including one minute of initial explanation, making it 38 min shorter than the PPA protocol.

Given the ease of use of this type of system to conduct clinical assessments, previous studies have developed and validated similar methods to assess fall risk in healthy and pathological populations [48,49,50]. However, of those used in the older adult population, less than half evaluated the validity, and less than 25% assessed the reliability of their applications [48]. Similar to the device used in this study, the smartphone systems tested in healthy older adult populations are based on the performance of different functional tests, such as the sit-to-stand test (with variables of time, peak force, rate of force development, and peak power [28]), balance test (with variables of root mean square, peak accelerations, mean sway area, median power of signal frequency, and total power of signal frequency [51,52,53]), or TUG test (with variables of total time, jerk and maximum acceleration during sit to stand, step time, interstride trunk autocorrelation, root mean square, and duration for separate elements [54,55]). The innovative contribution of the present study, compared with previous work, is that it provides the most accurate classification model to predict different levels of fall risk among parametric and nonparametric models, these last using neural networks and raw signal data obtained from a functional test (i.e., TUG). Furthermore, our system allows for the classification of fall risk into six categories aligned with PPA.

Aiming to be more faithful to the reality of the daily activities of older people and to increase the external validity of the study, different assessment approaches can be considered. Previous studies have used smartphones for the long-term monitoring of people’s activity [44], including commercial systems for detecting sudden falls and generating alerts when falls occur [56]. These systems are intended for use in continuous monitoring and evaluation. The presented study offers a different approach: a quick and easy instrumented functional assessment that can be performed in a clinical setting, such as the TUG or the six-minute walk test. Future research should explore the feasibility of conducting the assessment with the smartphone placed in the trouser pocket, opening the possibility for continuous and real-world monitoring and evaluation of daily life activities.

Time saving, simplicity, and usability of the sensor tool and protocol are the major advantages of the proposed method, allowing its use in a clinical setting. This is a good start for future studies, which could focus on determining which part of the modified TUG indicates the main cause of the risk of falling, allowing clinicians to individualise the fall prevention programme. Moreover, it is known that continued registration can provide accurate data about fall risks through real daily life activities [57]. This is why the device used in this study evaluates a functional test (i.e., modified TUG), but in the end, it is still evaluated in a laboratory. Therefore, future studies would be interesting to assess the performance of this device in real-life activities.

## 5. Conclusions

The fall risk assessment procedure developed in this study showed fair to excellent reliability. It allowed the classification of fall risk severity into two categories (fallers and nonfallers) and six ordinal categories according to PPA. When parametric and nonparametric models of classification were compared, those in which neural networks were used showed the highest accuracy. The proposed method has potential as a useful screening tool that can be used in general clinical practice because of its simplicity, usefulness, and quickness.

## Figures and Tables

**Figure 1 sensors-23-06567-f001:**
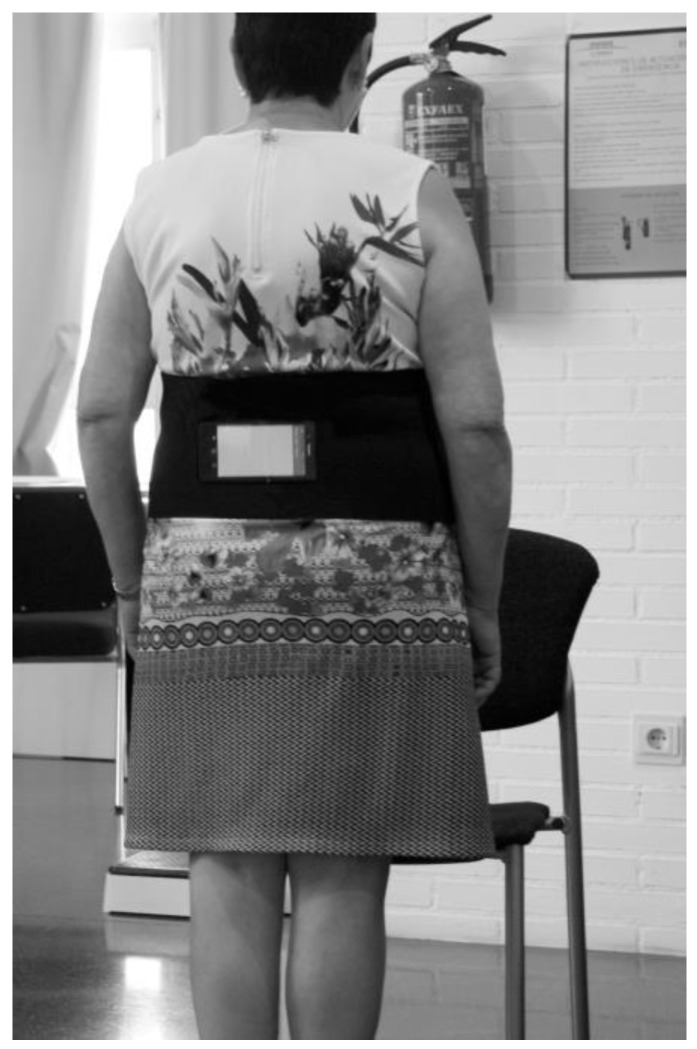
Participant FallSkip instrumentation.

**Figure 2 sensors-23-06567-f002:**
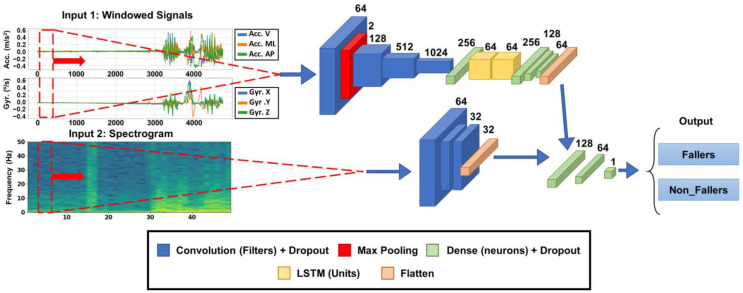
Structure of the fall risk level classification model with mixed input data. The temporal input datum (upper branch) is a moving window of 64 timestamps with the three axes of each sensor (accelerometer and gyroscope). This branch of the model comprises a series of convolutional and dropout layers and LSTM to automatically extract the temporal characteristics of the signals. The branch with the frequency information (lower branch) is the spectrogram image of the temporal signal. This branch comprises convolutional layers to extract information from the images. Acc.: accelerometer signal; Gyr.: Gyroscope signal; V: vertical; ML: medial-lateral; AP: anterior-posterior. This figure was self-made, created by Adobe InDesign^®^ 2023 (version 18.4).

**Table 2 sensors-23-06567-t002:** Comparison of the results of training and testing of classification models for two and six levels of fall risk.

Model	2 Fall Risk Levels (i.e., Fallers and Nonfallers)					6 Fall Risk Levels (i.e., Very Low, Low, Mild, Moderate, High, and Very High)				
	acc_Train	10fcv	acc Test	G_Mean	F1-Score	acc_Train	10fcv	acc Test	G_Mean	F1-Score
Logistic Regression	0.838	0.786	0.692	0.673	0.69	0.623	0.409	0.41	0.617	0.41
Ridge Classifier	0.818	0.753	0.692	0.663	0.69	0.5	0.364	0.333	0.482	0.33
LASSO	0.838	0.773	0.718	0.704	0.72	0.565	0.364	0.333	0.508	0.33
K-nearest Neighbours	0.838	0.721	0.641	0.629	0.64	0.623	0.364	0.333	0.535	0.33
Naive Bayes	0.76	0.753	0.692	0.682	0.69	0.468	0.331	0.385	0.602	0.38
Linear Discriminant Analysis	0.818	0.753	0.692	0.663	0.69	0.597	0.396	0.436	0.626	0.44
Decision Tree	1	0.643	0.667	0.67	0.67	1	0.331	0.436	0.631	0.44
Perceptron	0.76	0.746	0.667	0.651	0.67	0.435	0.337	0.282	0.483	0.28
Multilayer Perceptron	1	0.76	0.667	0.651	0.67	1	0.422	0.462	0.657	0.46
Stochastic Gradient Descent	0.805	0.74	0.641	0.629	0.64	0.526	0.382	0.41	0.594	0.41
Gradient Boosting	1	0.772	0.641	0.629	0.64	1	0.434	0.41	0.639	0.41
XGBoost	1	0.721	0.667	0.67	0.67	1	0.363	0.385	0.628	0.38
Support Vector Machine	0.903	0.74	0.615	0.598	0.62	0.662	0.402	0.359	0.491	0.36
Random Forest	1	0.798	0.615	0.598	0.62	1	0.429	0.385	0.584	0.38
AdaBoost	1	0.772	0.564	0.564	0.56	0.494	0.389	0.205	0.333	0.21
Multi-head CNN+LSTM	1	0.991	1	1	1	1	0.987	1	1	1

ICC—Intraclass Correlation Index of the three repetitions of the test; MLDisp—range of the medial–lateral displacement of the centre of mass (CoM) in balance phase; APDisp—range of the anterior–posterior displacement of the CoM in balance phase; VRange—range of vertical displacement of the CoM in gait phase; MLRange—range of medial–lateral displacement of the CoM in gait phase; PTurnSit—power of turn to sit; PStand—power to sit to stand; APJerkSit—anterior–posterior jerk to sit; APJerkStand—anterior–posterior jerk to sit and to stand.

## Data Availability

Not applicable.

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
