# Peer review of "The Reliability and Accuracy of a Fall Risk Assessment Procedure Using Mobile Smartphone Sensors Compared with a Physiological Profile Assessment"

_sensors, 2023, doi:10.3390/s23146567_

Round 1

Author Response

REVIEWER 1

The authors would like to thank the reviewer for the valuable suggestions and the appreciations

words about our research. We hope that our changes, following your reviews, could improve the quality of the manuscript.

A brief summary:

This study designs a fall risk assessment program using mobile smartphone sensors. The aim of this study was to evaluate the reliability and accuracy of an easy-to-use fall risk assessment based on a smartphone and validate it comparing it to the Physiological Profile Assessment (PPA) results. The data of the Timed-Up and Go test with the smartphone sensors for sixty-five (or thirty-six) participants older than 55 years were analyzed. Different classifiers models, including parametric models and non-parametric ones with neural networks, were used to classify fallers from non-fallers.

General concept comments:

The innovation of the study is insufficient with loose writing style. Since this manuscript was submitted to the special issue titled "Sensor Technology for Improving Human Movements and Postures: Part 11", how to utilize the biomechanical sensors and the acquired data during gait is of the readers' concern. The authors mentioned that "The mobility assessment was performed by a Java App running on a smartphone (Xiaomi Redmi 4x Model MAG138) with a 3D inertial sensor (High­ Performance 6-Axis MEMS Motion Tracking) composed of 3-axis gyroscope, 3-axis accelerometer and Digital Motion Processor (TDK-ICM-20689) at 100 Hz". Is the inertial sensor embedded in the smartphone?

We want to thank the reviewer for his/her comment. We have used a smartphone (Xiaomi) which includes a movement sensor embedded “6-Axis MEMS Motion Tracking” and the integrated movement digital processor “TDK-ICM-20689”, which are configured to record the accelerometer and gyro signals at 100 Hz. To clarify this information at the text, we have specified that the sensor is inserted in the smartphone, as suggested, in the “2.2 Experimental procedure” subsection.

This expression implicated that the data were collected at a fixed sampling rate of 100 Hz. However, linear interpolation was used to mitigate the non-constant device sample rate of the smartphone (in line 146).

The authors realize that the explanation in the manuscript was unclear and should be corrected. The data were, indeed, collected at a fixed sampling rate of 100 Hz. However, when acquiring the signal from a sensor at a constant frequency of 100 Hz, it is common practice to employ linear interpolation [1]. This technique serves as a preventive measure to avoid the loss of samples during the data acquisition process. By interpolating the signal, we ensure that the recorded data points are evenly distributed in time and that no information is missing between consecutive samples. Although no sample loss has been observed in the tests conducted, following the recommendations from relevant literature, such as configuring the Android system to minimize delays, helps maintain the integrity of the acquired data.

We have rewritten the "2.3 Sensor data analysis" subsection and we have also provided a new citation to better explain this process.

“Data were collected at a fixed sampling rate of 100 HZ from the smartphone embedded sensors and the raw signals were analyzed as in a previous study of our research group [22,26–28]. An interpolation of the recorded signal was performed to ensure that the data points were evenly distributed in time and that no information was missing between consecutive samples”

[1] S. Nishiguchi, M. Yamada, K. Nagai, S. Mori, Y. Kajiwara, T. Sonoda, K. Yoshimura, H. Yoshitomi, H. Ito, K. Okamoto, T. Ito, S. Muto, T. Ishihara, T. Aoyama Reliability and Validity of Gait Analysis by Android

 From the viewer's perspective, the approximation of acceleration data would affect the accuracy of computation. The viewer strongly suggests that the authors could take the combination of individual inertial sensor and smartphone into consideration. For instance, the IMUs (MTx, Xsens, Netherlands) were used in the reference [25] in the manuscript.

The authors agree with the reviewer about the possible loss of accuracy in the computation of the COM excursions using the smartphone embedded inertial sensor. The reviewer’s proposal of using a second sensor to combine the measurements to achieve more accurate results would probably increase the sensitivity of the system. Nevertheless, one of the interests of the study was to evaluate the performance of a system that meets the requirements of the clinical context, in time-consumption and simplicity of use.

The possibility of using the smartphone as the recording system and also as the interface to perform the assessment and report the results was considered as very user friendly, perfectly matching with the clinical needs. That's the reason why the goal of this manuscript was to test the sensitivity of the system, as it was, to detect risk of falls.

However, the reviewer has remarked an important consideration, the possibility to combine individual and smartphone sensors. We are sure that this combination could provide more detail about movement, but on the contrary, could jeopardize the simplicity of the clinical protocol to be used in the clinical context, so we have included this as a limitation.

The authors realize that these issues were not specified in the text and some clarifications have been added.

Introduction section:

“Instrumented assessment using the sensors embedded in a smartphone device may have several advantages when used in the clinical setting. In contrast to the simplicity of the assessment using a smartphone, other devices (e.g., Kinect, Doppler radar, or multimodal strategies) may require specific hardware setups, dedicated environments, or qualified personnel to operate the systems or perform the analysis.”

Discussion section:

“It would also be possible to use a combination of different devices for the registering (as having more than one inertial sensor) of different input data typologies, (i.e., raw signals from the sensor or the spectrogram), even including clinical variables, which may allow a more complete approach [42]. However, this may impact on the usability and speed of the assessments which are essential in clinical context.”

Moreover, the authors trained a customized CNN for the classification of fallers and non-fallers. The proposed neural network-based classifier showed an accuracy of 0.99 and 0.98, respectively. Since the sample size (65*3) is too small in in machine learning, overfitting should be taken seriously.

 We agree with the reviewer, and share their concern about the generalizability of the models. For that reason, we have taken safeguards to avoid overfitting, such us including dropout layers (see next answer) and, in addition to using a repeated stratified 10-fold validation, we took a 20% out of the sample for testing the accuracy of the models with data that had not been used in any stage of the training of the model of its hyperparameters, as indicated in the paper.

Nevertheless, the reviewers are right in pointing out that the remarkably high accuracies that we obtained might be overoptimistic, influenced by the small size of the sample, and the limited diversity of health conditions. This has been pointed out in two points of the discussion:

  • When comparing our model with that reported in [48], we have commented:

One of those studies has demonstrated that neural network analysis is able to classify older adults as a fallers and non-fallers using sensor data based on TUG test, among people with different health conditions such as cognitive and physical impairments. The size and diversity of our sample was smaller, and therefore it might not be sufficiently representative to generalize the results, although it is a good starting point for future studies with a larger sample size, including fall risk classification for specific pathologies as in previous studies

  • And after discussing the 99% and 98% accuracies:

Such remarkable results should be appraised with caution, due to the relatively small sample size. Although the method followed in this study included safeguards to avoid overfitting of the model, such as using dropout layers in the neural network, and sparing 20% of the sample for independent testing, the limited size of the whole sample implied that the validation and test sets had less than 40 subjects, with subgroups that had less than 5 cases in the worst case. Even perfect fits observed with such small samples, as happened in this study, might be obtained by chance with models whose real accuracy is much smaller.

From the structure of neural network (Figure 3 in the manuscript), the viewer did not find any dropout layer, which can reduce overfitting.

We want to thank the reviewer since his/her comment has made us realize of the lack of explanation about it in our manuscript. The model actually includes dropout layers, which randomly set input units to 0 in order to prevent overfitting. This have been properly explained in section 2.5.2 of the text and also illustrated in figure 3.

Minor comments:

The minor comments are as follows.

  1. In lines 98 (section 2.3) and 281 (section 3.1), the authors mentioned that thirty-six participants were recruited for the study. However, in the abstract section (in line 19) and section 2.5.3 (in line 268), data of 65 participants were used for analysis. This makes the viewers feel confused.

We apologized for this mistake. We included 65 participants and we have corrected it throughout the text.

  1. For the inferential analysis, the intra-class correlation coefficient for the average of random measures, ICC(l,k), was calculated to assess the reliability of the eight variables of the test. In this paper, the selection of k also concerned the viewer. Please read the paper [l] as below, and figure out the number of k which should be appropriate for the case in this paper (test-retest reliability).

We would like to apologize since we actually conducted the reliability analysis using an ICC (2,1), as the assessments were conducted once. We have corrected in the text.

  1. In table 2, is "acc_text" associated with the accuracy during the test process? If so, it should be written as "acc test".

The term has been reworded as suggested.

  1. In this paper, t-tests were used. were all the variables normal distributed or not?

 Both homoscedasticity and normality were assumed for the variables. We have clarified it at results section, at the participants subsection.

  1. The authors mentioned that the kinematic parameters during gait were calculated from the inertial sensors. Please introduce the algorithm in more detail in the manuscript.

Following the suggestion, we have added more information regarding the algorithm used, in the following paragraphs, which explain how we obtained the angles and positions that were used for segmenting the phases of the test and calculating the spatio-temporal parameters, respectively:

The process of calculating the dependent variables started with the identification of the sensor’s orientation from the accelerations and angular velocities using Favre’s method [30]. These orientations were expressed in Euler angles (Roll, Pitch and Yaw), which were used during the segmentation process of the test phases (balance, gait, turn to sit, sit to stand and gait), by identifying the instants where changes of direction (turn to sit) occurred.

Then the position of the sensor was calculated by the double integration of the accel-eration signal. This was done analytically in the frequency domain, using the Fourier transform and its inverse as in [31], in order to avoid cumulative drift due to the integra-tion process

.The references cited in those paragraphs are:

References:

[30] Favre, J.; Jolles, B.M.; Siegrist, O.; Aminian, K. Quaternion-Based Fusion of Gyroscopes and Accelerometers to Improve 3D Angle Measurement. Electron. Lett 2006, 42, 612–614.

[31] Ribeiro, J.G.T.; Castro, J.T.P.D.; Freire, J.L.F. Using the Fft- Ddi Method to Measure Displacements with Piezoelectric, Resis-tive and Icp Accelerometers.; February 2003.

Reference:

[1] Thiese MS, Arnold ZC, Walker SD. The misuse and abuse of statistics in biomedical research. Biochem Med (Zagreb). 2015;25(1):5-11.

Reviewer 2 Report

Important paper, still need to be improved before publication:

0.       Emphasize more the contribution of your work compared to other works with cellphone and fall detection, in particular your contribution compare to your previous work e.g. [18]

1.       Reference – too short .Missing in the paper in the introduction a pargaprh that dedicated to standard technology to detect falls: Kinect, radar based, sonar based, and video based. For example, write few of of the following papers:

Blumrosen, Gaddi, et al. "A real-time kinect signature-based patient home monitoring system." Sensors 16.11 (2016): 1965.

Amin, Moeness G., et al. "Radar signal processing for elderly fall detection: The future for in-home monitoring." IEEE Signal Processing Magazine 33.2 (2016): 71-80.

Espinosa, Ricardo, et al. "A vision-based approach for fall detection using multiple cameras and convolutional neural networks: A case study using the UP-Fall detection dataset." Computers in biology and medicine 115 (2019): 103520.

Tsai, Tsung-Han, and Chin-Wei Hsu. "Implementation of fall detection system based on 3D skeleton for deep learning technique." IEEE Access 7 (2019): 153049-153059.

2.       Writing wise – do not use bullets, but write part of paragraph (can include numbers, and ; )

3.       The (paramteric) features you use – please give references to how you got them and why you don’t use others

4.       Define what is 2 and 6 fall risk fallers – not clear

5.       Emphasize more why your results are better for 2 fall risk faller, and not also for the 6 fall risk faller – not clear the terms. I would assume that 6 fall risk faller (higher risk?) is easier to predict

6.       Speak about how you can combine parametric + non parametric

7.       Discuss if you can use the phone in the pocket, or through daily life activities

english is reltively fine and clear

Author Response

REVIEWER 2

The authors would like to thank the reviewer for his/her appreciations and the valuable comments. We hope that they have been convincingly resolved.

  1. Emphasize more the contribution of your work compared to other works with cellphone and fall detection, in particular your contribution compares to your previous work e.g. [18]

Thank you for your suggestion. We have extended this topic at the discussion section, also comparing with similar studies.

  1. Reference – too short. Missing in the paper in the introduction a pargaprh that dedicated to standard technology to detect falls: Kinect, radar based, sonar based, and video based. For example, write few of of the following papers:

Blumrosen, Gaddi, et al. "A real-time kinect signature-based patient home monitoring system." Sensors 16.11 (2016): 1965.

Amin, Moeness G., et al. "Radar signal processing for elderly fall detection: The future for in-home monitoring." IEEE Signal Processing Magazine 33.2 (2016): 71-80.

Espinosa, Ricardo, et al. "A vision-based approach for fall detection using multiple cameras and convolutional neural networks: A case study using the UP-Fall detection dataset." Computers in biology and medicine 115 (2019): 103520.

Tsai, Tsung-Han, and Chin-Wei Hsu. "Implementation of fall detection system based on 3D skeleton for deep learning technique." IEEE Access 7 (2019): 153049-153059.

Thank you for your appreciation. We have added a paragraph talking about technology to detect falls at the introduction section and we have compared it to smartphone used in this study.

  1. Writing wise – do not use bullets, but write part of paragraph (can include numbers, and ; )

We have removed all bullets and we put the information into paragraphs including an enumeration.

  1. The (paramteric) features you use – please give references to how you got them and why you don’t use others

Thank you for your suggestion. We have included more information regarding procedure. Concretely:

Section 2.3 The process of obtaining the different variables was to obtain the orientation of the sensor from the accelerations and angular velocities using the Favre [3] method. These orientations are expressed in both Euler angles (Roll, Pitch and Yaw) and quaternions (qw, qx, qy, qz). The orientation is used during the process of delineating the phases of the test by identifying the instants where changes of direction (turn to sit) occur.

To obtain the Position in the three axes of movement from the accelerations. The process of obtaining this magnitude is associated with a process of integration (cumulative sum), which causes a drift in the calculation. For this reason, it was decided to apply a calculation method that performs a double integration using the Fourier transform and its inverse [4]. This process performs an analytical integration of a signal in the frequency domain, in our case the acceleration. Once the position has been obtained, the variables for each of the phases of the test are obtained mentioned in the text.

Section 2.2. The reason for instrumenting the lumbar region is because it is the easiest area to place a sensor that moves with the body and is close to the body's center of masses. Furthermore, previous studies have determined that the kinematics of the lower lumbar region provide information on energy consumption during gait [5].

References:

[3] Favre, J.; Jolles, B.M.; Siegrist, O.; Aminian, K. Quaternion-Based Fusion of Gyroscopes and Accelerometers to Improve 3D Angle Measurement. Electron. Lett 2006, 42, 612–614.

[4] Ribeiro, J.G.T.; Castro, J.T.P.D.; Freire, J.L.F. Using the Fft- Ddi Method to Measure Displacements with Piezoelectric, Resis-tive and Icp Accelerometers.; February 2003.

[5] Kerrigan, D.C.; Thirunarayan, M.A.; Duff-Raffaele, M. The Vertical Displacement of the Center of Mass during Walking: A Comparison of Four Measurement Methods. 1998

  1. Define what is 2 and 6 fall risk fallers – not clear

2 fall risk level means that the classification only included 2 levels of fall risk (fallers and non-fallers). 6 fall risk levels means that 6 levels according to PPA scale were included in the classification (i.e., very low risk, low risk, mild risk, moderate risk, high risk, and very high risk). As we understand that this may be confusing for the reader, we have added the punctuation for each level at “Classification models” subsection. Moreover, we have defined 2-degrees scale and 6-degrees scale at Table 2. We hope that this will make the information clearer.

  1. Emphasize more why your results are better for 2 fall risk faller, and not also for the 6 fall risk faller – not clear the terms. I would assume that 6 fall risk faller (higher risk?) is easier to predict 

We believe that now that we address the anterior comment, specifying the meaning of the 2 or 6 levels of classification, and considering that differences are minimum, this issue is resolved.  Thank you for highlighting this issue.

  1. Speak about how you can combine parametric + non parametric

Following your suggestion, we have included the following information:

A different neural network topology could be used to combine parametric and non-parametric models. It would be necessary to include a third input branch where the computed variables (APDisp, MLDisp, VRange, etc.) would be included using Densas layers and then concatenate them at the same junction point of the other two input branches. We have used this approach in a previous study where we classified the first three levels of Parkinson's disease [2]. For this study it was not necessary to use it as the results obtained were good enough to increase the complexity of the model and its subsequent training time.

[2] Pedrero-Sánchez, J.F.; Belda-Lois, J.M.; Serra-Añó, P.; Mollà-Casanova, S.; López-Pascual, J. Classification of Parkinson’s Dis-ease Stages with a Two-Stage Deep Neural Network. Frontiers in Aging Neuroscience 2023, 15

  1. Discuss if you can use the phone in the pocket, or through daily life activities

Thank you for your comment, this would be an interesting approach, we have added it at the end of the discussion. There are previous studies that use a smartphone to monitor people's activity [6], including commercial systems to detect sudden falls and generate alerts when a fall occurs [7]. These systems are intended to be used in a monitoring and continuous evaluation environment. The presented study has a different approach; it is a quick and easy instrumented functional assessment that can be performed in a clinical setting, such as the TUG or the six-minute walk test. To use the smartphone and the proposed model as a monitoring method by carrying the smartphone in the pocket, it would be necessary to generate a database with this type of measurements and train the model since the generated signals would have a different morphology.

References:

[6] Nait Aicha, A.; Englebienne, G.; van Schooten, K.S.; Pijnappels, M.; Kröse, B. Deep Learning to Predict Falls in Older Adults Based on Daily-Life Trunk Accelerometry. Sensors (Basel, Switzerland) 2018, 18, 1654, doi:10.3390/s18051654.

[7] Williams, G.; Doughty, K.; Cameron, K.; Bradley, D.A. A Smart Fall and Activity Monitor for Telecare Applications. In Pro-ceedings of the Proceedings of the 20th Annual International Conference of the IEEE Engineering in Medicine and Biology Society, 1998; October 1998; Vol. 3, pp. 1151–1154 vol.3.

Round 2

Reviewer 1 Report

The revised manuscript  is considered to be published in Sensors. 

The English language is appropriate and understandable. 

Author Response

Thank you very much.

Reviewer 2 Report

1. Still the intro too short, and today many of the methods to detect fall are using in house video + sonar+ radar, and pity you don't write pargraph about general methods to detect falls, then write the you focus on subset of inertial sensors on phone

2. some figures are not needed, and taken with out giving credit - like fig 2, and 3

3. combining paramtric and data driven can give additional computational gain - please show what is the gain - the training time, is less an issue

Author Response

The authors would like to thank the reviewer for the valuable suggestions and the appreciations. Please, see below manuscript changes following your reviews We hope this could improve the quality of the manuscript.

  1. Still the intro too short, and today many of the methods to detect fall are using in house video + sonar+ radar, and pity you don't write pargraph about general methods to detect falls, then write the you focus on subset of inertial sensors on phone.

Thank you for your suggestion. We have added two paragraphs. In the first one, we have described different sensors used for fall risk assessment, including pulse-Doppler radar, Kinect, ultrasound, optical Time-of-Flight and web camaras, and their advantages. In the second included paragraph we have cited other devices as force and pressure sensors, and inertial sensors, focusing on the last one. Moreover, we have added more bibliography regarding these different sensors:

Argañarás, J.G.; Wong, Y.T.; Begg, R.; Karmakar, N.C. State-of-the-Art Wearable Sensors and Possibilities for Radar in Fall Prevention. Sensors (Basel) 2021, 21, 6836, doi:10.3390/s21206836.

Mehmood, A.; Sabatier, J.M.; Bradley, M.; Ekimov, A. Extraction of the Velocity of Walking Human’s Body Segments Using Ultrasonic Doppler. J Acoust Soc Am 2010, 128, EL316-322, doi:10.1121/1.3501115.

Phillips, C.E.; Keller, J.; Popescu, M.; Skubic, M.; Rantz, M.J.; Cuddihy, P.E.; Yardibi, T. Radar Walk Detection in the Apartments of Elderly. Annu Int Conf IEEE Eng Med Biol Soc 2012, 2012, 5863–5866, doi:10.1109/EMBC.2012.6347327.

Wang, F.; Stone, E.; Skubic, M.; Keller, J.M.; Abbott, C.; Rantz, M. Toward a Passive Low-Cost in-Home Gait Assessment System for Older Adults. IEEE J Biomed Health Inform 2013, 17, 346–355, doi:10.1109/JBHI.2012.2233745.

Wang, S.; Skubic, M.; Zhu, Y. Activity Density Map Visualization and Dissimilarity Comparison for Eldercare Monitoring. IEEE Trans Inf Technol Biomed 2012, 16, 607–614, doi:10.1109/TITB.2012.2196439.

  1. some figures are not needed, and taken with out giving credit - like fig 2, and 3

We have deleted fig 2 since similar figures are included in previous research papers of our group and the steps of the assessment procedure are explained in text (we have also referenced our previous works). Regarding figure 3, we cannot delete it because the reviewer 1 asked us to expand the information of the figure in the previous round of reviewing.

  1. combining paramtric and data driven can give additional computational gain - please show what is the gain - the training time, is less an issue

Thank you for your comment. We believe that it would be feasible to increase the model performance with a more sophisticated design. As shown in a previous manuscript [1], a mixed input model comprising three types of data (biomechanical variables, time domain and frequency domain) outperformed the accuracy of simpler models in the classification of Parkinson’s Disease stages. In that study a sensitivity analysis was done, remarking the relevant contribution of all the model input data types. The application of such two-stage model would be out of the scope of the present manuscript, since the accuracy of the model used in this experiment was already almost 100%. However, we believe that it would be convenient to test this approach in futures studies with larger samples.

To better highlight the importance of considering different data inputs in the classification models, we have added this paragraph in the discussion section:

“Future studies with larger samples may also test the inclusion of clinical variables as a third independent branch of the model, since this approach has shown very promising accuracy for the classification of Parkinson’s disease stages [47].”

[1] Pedrero-Sánchez, J.F.; Belda-Lois, J.M.; Serra-Añó, P.; Mollà-Casanova, S.; López-Pascual, J. Classification of Parkinson’s Dis-ease Stages with a Two-Stage Deep Neural Network. Frontiers in Aging Neuroscience 2023, 15
